# Contrastive Multiple Instance Learning: An Unsupervised Framework for Learning Slide-Level Representations of Whole Slide Histopathology Images without Labels

**DOI:** 10.3390/cancers14235778

**Published:** 2022-11-24

**Authors:** Thomas E. Tavolara, Metin N. Gurcan, M. Khalid Khan Niazi

**Affiliations:** Center for Biomedical Informatics, Wake Forest School of Medicine, Winston-Salem, NC 27101, USA

**Keywords:** deep learning, histopathology, self-supervised, contrastive, multiple instance learning

## Abstract

**Simple Summary:**

Recent AI methods in the automated analysis of histopathological imaging data associated with cancer have trended towards less supervision by humans. Yet, there are circumstances when humans cannot lend a hand to AI. Hence, we present an unsupervised method to learn meaningful features from histopathological imaging data. We applied our method to non-small cell lung cancer subtyping as a classification prototype and breast cancer proliferation scoring as a regression prototype. Our AI method achieves high accuracy and correlation, respectively. Additional experiments aimed at reducing the amount of available data demonstrated that the learned features are robust. Overall, our AI method approaches the analysis of histopathological imaging data in a novel manner, where meaningful features can be learned from without the need for any supervision my humans. The proposed method stands to benefit the field, as it theoretically enables researchers to benefit from completely raw histopathology imaging data.

**Abstract:**

Recent methods in computational pathology have trended towards semi- and weakly-supervised methods requiring only slide-level labels. Yet, even slide-level labels may be absent or irrelevant to the application of interest, such as in clinical trials. Hence, we present a fully unsupervised method to learn meaningful, compact representations of WSIs. Our method initially trains a tile-wise encoder using SimCLR, from which subsets of tile-wise embeddings are extracted and fused via an attention-based multiple-instance learning framework to yield slide-level representations. The resulting set of intra-slide-level and inter-slide-level embeddings are attracted and repelled via contrastive loss, respectively. This resulted in slide-level representations with self-supervision. We applied our method to two tasks— (1) non-small cell lung cancer subtyping (NSCLC) as a classification prototype and (2) breast cancer proliferation scoring (TUPAC16) as a regression prototype—and achieved an AUC of 0.8641 ± 0.0115 and correlation (R^2^) of 0.5740 ± 0.0970, respectively. Ablation experiments demonstrate that the resulting unsupervised slide-level feature space can be fine-tuned with small datasets for both tasks. Overall, our method approaches computational pathology in a novel manner, where meaningful features can be learned from whole-slide images without the need for annotations of slide-level labels. The proposed method stands to benefit computational pathology, as it theoretically enables researchers to benefit from completely unlabeled whole-slide images.

## 1. Introduction

Histopathological analyses play a central role in the characterization of biological tissues. Increasingly, whole-slide imaging (WSI) of tissues, in tandem with inexpensive storage and fast networks for data transfer, has made it possible to curate large databases of digitized tissue sections [1]. Furthermore, advances in deep learning methods have enabled scientists to develop automated histopathological analysis methods on whole-slide images, ranging from primitive applications such as in nuclei detection [2] and in mitosis detection [3] to more advanced applications, such as tumor grading [4].

Despite successful application to various diagnostic and prognostic problems [1,5], developing methods for computational pathology perpetually rely on painstakingly annotated nuclei, cells, and tissue structures [6,7,8,9,10]. This is driven primarily by the prevalence of annotation-heavy supervised methods in more generalized computer vision applications [11,12,13]. Unlike general computer vision applications and other medical imaging modalities [14], reliance on annotations heavily limits research in computational pathology, as annotations must be performed by expert pathologists [15]. Furthermore, annotations are labor-intensive and often subject to significant inter- and intra- reader variability [1]. Finally, medical image datasets are vanishingly small compared to general-purpose computer vision datasets [14,16]. It is no wonder that recent high-profile publications in computational pathology have moved away from fully-supervised methods to semi- and weakly-supervised methods [17,18,19,20].

In continuance with recent trends towards less supervision, our goal is to develop an unsupervised method to learn meaningful, compact representations of WSIs. Such methods exist in other medical imaging modalities [14] but to our knowledge, no such method currently exists for computational pathology. However, several unsupervised (specifically, self-supervised) methods exist to learn *patch-wise* representations within computational pathology. These works define pretext tasks from which patch-wise feature representations are learned. Such pretext tasks include contrastive predictive coding [21], contrastive learning on adjacent image patches [22], contrastive learning using SimCLR [23,24,25], and SimSiam [26] with an additional stop-gradient for adjacent patches [27]. Many methods utilize generic features derived from ImageNet as their patch-wise feature representations [17,18,28]. Neural image compression [29] compares several self-supervised pretext training tasks to create feature-rich WSI representations but is impeded by dimensionality [30], due to the sheer number of parameters associated with each compressed slide. Finally, several autoencoder and derivative methods have been applied to learn compact patch-wise representations without labels to a variety of application areas, including nuclei detection [31,32], cell detection and classification [33], drug efficacy prediction [34], tumor subtype classification [35], and lymph node metastases detection [36].

By and large, these studies have three main findings. Firstly, self-supervised pretext tasks for learning patch-wise features tend to outperform ImageNet features. This is probably because ImageNet features are generic with respect to everyday objects, whereas features derived from self-supervision are purely based on WSI patches. Furthermore, the transformation space can be tweaked to suit pathology (i.e., scale invariance). Secondly, there is a saturation point, at which time adding more patches to the pretext tasks does not offer downstream performance gains. This result is not reflected in general-purpose self-supervised learning, where more images result in higher downstream performance. Stacke et al. [23] propose that this may be due to the redundancy in WSI patches (i.e., anatomies tend to be repeated). Thirdly, fewer labels are required when compared to training from scratch or transfer learning from ImageNet weights. This is also reflected in general-purpose applications.

Due to these promising empirical findings and theoretical perspectives for patch-wise classification of histopathology images, a few studies have combined self-supervised patch-level representations with weakly supervised multiple instance learning (MIL) [37,38] methods (i.e., for WSI analysis). Lu et al. [21] utilized contrastive predictive coding on WSI patches as a pre-training step for subsequent MIL-based classification of breast cancer WSIs. Another study by Li et al. [24] utilized SimCLR [39] on WSI patches as a pre-training step for subsequent MIL-based classification. Fashi et al. [40] utilized contrastive learning based on site-of-origin labels as pseudo-labels for pre-training then applied attention pooling on the resulting embeddings to classify WSIs. All studies demonstrated similar benefits at the slide level that were considered at the patch level.

Despite promising research showing the benefits of self-supervision and MIL, one aspect of WSI analysis that has yet to be addressed in the literature is whether self-supervision can be applied at the slide level via multiple instance learning. Thus, we propose a novel fusion of self-supervision and MIL, which we call SS-MIL, as a method for learning *unsupervised* representations for WSIs. Our method first trains a patch-wise encoder using SimCLR [39]. Then, each patch is embedded to yield a collection of instances for each WSI. Each collection is divided into multiple subsets, each representing the WSI, which MIL then fuses to yield multiple slide-level representations. Through this operation on the same slide, a positive pair for contrastive learning is created. Similarly, a negative pair is created for a different slide. This MIL model is then trained using contrastive loss. The resulting slide-level representations are therefore created without any supervision. We then apply supervision to these unsupervised representations in both a classification and regression task. For classification, we subtype NSCLC into lung adenocarcinoma (LUAD, *n* = 541) and squamous cell carcinoma (LUSC, *n* = 512) using the publicly available Cancer Genome Atlas (TCGA)-NSCLC dataset. For regression, we score the degree of proliferation (i.e., mitotic activity) in breast cancer using the publicly available TUPAC16 dataset. We demonstrate through ablation experiments that the unsupervised slide-level feature space can easily be fine-tuned using a fraction of labeled slides, indicating the unsupervised feature space is meaningful. Not only is SS-MIL a novel, label-less approach to computational pathology, but it also creates an opportunity in which vast amounts of unlabeled or irrelevantly labeled WSIs may benefit the development of models in computational pathology.

Our main contributions in this paper are: (1) a novel fusion of self-supervision at the slidelevel; (2) an unsupervised method (SS-MIL) to learn representations of WSIs; (3) empirical evidence suggesting that ImageNet features may *outperform* self-supervised features at the slide-level, contrary to previous studies; and (4) empirical evidence via ablation studies demonstrating that the resulting unsupervised feature space is rich.

## 2. Related Work

### 2.1. Self-Supervised Learning

Self-supervised learning is a subclass of unsupervised learning in which feature-rich latent space representations are obtained without the need for manual labels. Rather, the objective is altered such that it relies only on the data itself. Early deep learning approaches focused primarily on generative approaches in which representations are learned as a byproduct of image reconstruction from the learned latent space [41]. For example, one may add noise to an image, train an autoencoder [42] to reconstruct the image, and then utilize the learned latent space as a representation of the input image. Such methods were challenged and matched by very popular GAN-based [43] approaches which used adversarial (and sometimes reconstruction-based) objectives [44,45,46]. Alternatively, pseudo-labels can be artificially created from the data directly. Such pseudo-labels have traditionally been generated by exploiting the spatial characteristics of images—including relative position, jigsaws, or rotation [47,48,49]. Such methods perturb the image using said transformations and then minimize an objective attempting to predict said parameter associated with said transformation.

More recently, contrastive learning approaches to self-supervised learning have become increasingly popular. These methods draw their inspiration from the perturbation aspect of self-supervision. Their key assumption is that the learned feature representations of any two random perturbations of the same image should be similar, colloquially called “positive pairs”. Conversely, the learned representations for two perturbed images *not* from the same image should not be the same, colloquially called “negative pairs”. Furthermore, the set of perturbations (or “transformations” colloquially) is expanded to include simple image augmentations (resizing, flipping, color jittering, etc.). An analogous effect is derived from data augmentation, in which several of the same transformations are applied in order to increase the model robustness [50]. Several popular implementations exist to learn effective transformation-invariant representations [26,39,51,52,53,54,55,56,57]. Other similarly-inspired methods do away with negative pairs and instead exploit a slightly different aspect of deep learning models to learn effective representations. These exploitable features include temperature-based loss of function, exponential moving averages of model weights in a teacher/student fashion, comparing prototype feature representations instead of raw latent representations, and sparsity [54,55,57,58]. These general-purpose self-supervised methods have shown that the resulting feature space is so rich that only 1% of the original training labels are required to train a classifier that rivals that trained with full supervision. Thus, self-supervised methods within computational pathology are of particular interest, as labels (or annotations) are expensive, time-consuming, and subject to inter-reader variability [59].

Several studies have applied patch-wise self-supervised methods to computational pathology. A large-scale study by Ciga et al. [60] applied SimCLR [39] to 57 distinct histopathology datasets for patch-wise classification, regression, and segmentation. They showed that self-supervised pre-training and then fine-tuning on a small, labeled dataset consistently performed on par with full supervision using ImageNet pre-training. Furthermore, they showed that their pre-trained model *outperformed* ImageNet when labels were limited. Moreover, they demonstrated that the number of patches utilized during pre-training has a saturation point, after which no additional downstream performance is obtained (~50,000). Finally, they showed that combining image patches from different sites did not improve downstream performance and was comparable to single-site performance. The same conclusions were drawn earlier by Stacke et al. [23], who applied SimCLR to patch-wise breast cancer and skin cancer classification. They also showed that the set of optimal transformations *changed* depending on the dataset utilized. Finally, Srinidhi et al. [61] developed a custom contrastive learning-based self-supervised model based on resolution sequence ordering (rather than transformations). They showed that their histopathology-specific model outperformed a general-purpose contrastive learning self-supervised model (i.e., MoCo [58]) on three datasets—tumor metastasis detection, tissue type classification, and tumor cellularity quantification—under annotation-limited settings. Lastly, Wang et al. developed a self-supervised method combined with self-attention to learn the patch-level embeddings [62,63], and then performed slide-level image retrieval based on said embeddings [64].

### 2.2. Multiple Instance Learning (MIL)

MIL [37,38] is a machine learning method where labels are assigned to collections (called bags) rather than individual examples (called instances), as in conventional machine learning. For histopathological analyses, bags consist of tessellated WSIs, in which each tile is a small unannotated image sampled from the WSI. It is popular for WSI analyses, as only slide-level (i.e., weak) labels are required for its training and implementation, thus negating the need for tissue-level annotations.

Courtiol et al. [65] were the first to tackle WSI analysis with MIL. They applied their MIL method to a Camelyon16 [66], which consists of sentinel lymph nodes that are either present or absent in breast cancer metastases. Their method achieved a high area-under-the-curve (AUC) and attended specifically to pathologist-annotated metastases (rather than arbitrary tissue regions). This was notable at the time, as previous methods had relied on tediously annotated tissue ROIs. Several studies have since followed suit [4,17,20,25,67,68,69,70].

Metastasis detection is more difficult than tasks dealing with more generalized cancer presence or absence, in which the diseased tissue is larger and more diffuse. Campenella et al. [4] conducted the first study aiming to develop and apply an MIL method for this task. They applied their method to three large sets of WSIs, including an in-house prostate cancer dataset (*n* = 12,132), an external prostate cancer dataset (*n* = 12,727), and a skin cancer basal cell carcinoma dataset (*n* = 9962). Their results showed that MIL on large datasets outperforms strong supervision on small datasets. This was significant, as previous studies relied on pathologists’ annotations of tumor regions in small WSI datasets (with the obvious exception of earlier work by Courtiol et al.). Several studies have similarly applied MIL for tumor presence or absence classification in WSIs [34,35,36,37,38,39].

MIL has also been applied to subtyping tasks. Courtiol et al. [65] were the first to apply MIL to subtype cancer using WSIs—specifically, subtyping NSCLC into LUAD and LUSC using WSIs available from TCGA [71]—and drew similar conclusions and implications from their work with Camelyon16. The standard set by Courtiol et al. has endured through subsequent studies [17,25,68,69,70,72,73,74]. Wang et al. added an addition subtype for small-cell lung cancer (SCLC) [43] using an in-house dataset. Beyond lung cancer, other subtyping tasks have also been studied using MIL, most popularly renal-cell carcinoma subtyping into papillary, chromophobe, and clear cells [17,25,69]. Hashimoto et al. subtyped lymphomas into diffuse large B-cell (DLBCL) and non-DLBCL [75]. Lu et al. subtyped gliomas into glioblastoma, oligodendroglioma, and astrocytoma while ignoring oligoastrocytoma [76].

Finally, MIL has been applied to grading—most popularly, Gleason grading [77,78]. Bulten el al. [79] removed the need for manual annotations by utilizing two pre-trained models to (1) delineate a rough tumor outline and (2) remove epithelial tissue from WSIs. Then, all tissue regions were labeled with the pathologist’s reported Gleason pattern. During the training of their model, only “pure” biopsies were included (i.e., 3 + 3, 4 + 4, and 5 + 5). They showed that despite this weak labeling strategy, their method was able to accurately score Gleason grades outside the original domain (i.e., 3 + 4 and 4 + 3, etc.) and had a high consensus with experts. Subsequent studies have also applied MIL for Gleason grading [80,81], grading dysplasia of various cancers (i.e., normal, low-grade, and high-grade) [82,83], and grading of colorectal cancer (i.e., low-grade and high-grade) [84].

### 2.3. Self-Supervision and MIL

The majority of the MIL studied and mentioned thus far utilize models pre-trained on ImageNet as a feature extractor, in order to embed WSI patches into a feature space for subsequent utilization by some MIL algorithm. Given that some studies applying self-supervision to WSI patches showed improved performance over the ImageNet baseline, a natural question arises—why not use self-supervision for pre-training? Thus, several studies have followed suit. Lu et al. utilized contrastive predictive coding on WSI patches as a pre-training step for subsequent MIL-based classification of breast cancer WSIs [21]. Not only did their method outperform ImageNet pre-training, but they also showed that reducing the number of labels per class by 80% during MIL training did not significantly degrade performance. Another study by Li et al. [24] utilized SimCLR on WSI patches as a pre-training step for subsequent MIL-based classification. Alike Lu et al., they showed that these features outperformed ImageNet features on downstream MIL classification tasks. Furthermore, they showed the same increase in performance for multiple datasets and for several MIL methods (not just their own). Liu et al. [27] reached similar conclusions, albeit utilizing a slightly modified self-supervised objective which, in addition to exploiting transformation-invariance, incorporated locality-invariance. Rather remarkably, they noted that neighboring WSI patches look the same and thus should have similar embeddings. Finally, Fashi et al. utilized contrastive learning based on site-of-origin labels as pseudo-labels for pre-training, then applied attention pooling on the resulting embeddings to classify WSIs [40]. They astutely noted that site-of-origin is nearly always available and thus, should be incorporated into the self-supervised objective (i.e., as a label).

## 3. Methods

### 3.1. SimCLR for Effective Patch-Wise Representations

SimCLR [39] is a framework for learning effective representations of images without labels (i.e., self-supervised). The model consists of two parts: an encoder followed by a projection head. A pair of images is generated through random transformation methods (such as cropping, resizing, or color jittering) of an image from the dataset and fed into the network. This pair is known as a positive pair, as it is generated from the same source image. Conversely, a negative pair is a pair not generated from the same source image. Contrastive loss is then used to optimize the network. This loss function essentially rewards positive image pairs for which the outputs of the projection head are similar (in terms of cosine similarity) and penalizes negative image pairs for which the projection outputs are similar. In essence, the network learns embedded representations of input images invariant to their transformations. This is depicted in Figure 1.

Using this intuitively simple framework, the resulting encoder outputs representations of images that are easily discriminated by a single-layer multilayer perceptron (MLP) for classification purposes [39]. Their discriminative performance matches the performance of equivalent fully supervised methods. Several self-supervised methods inspired by SimCLR have consistently demonstrated that the pursuit of transformation invariance yields meaningful representations of images [18,54,55,85]. Self-supervised methods cannot be applied directly to WSIs, as the images are too large. Instead, self-supervision operates at the tile level. Several studies have carried such experiments out for tile-level classification [23,24,27,60]. This is justified by the observation that WSI tiles exhibit some of the same variation that SimCLR tries to achieve through transformation invariance. When a tile is sampled from a WSI—whether it be at different magnifications (scale), with slightly different staining (color), or with rotation—the learned representation of that tile should be the same. This is because regardless of scale, stain variations, and rotation, the information contained in that tile should be the same. Therefore, the same transformation variances that SimCLR attempts to minimize also apply to WSI tiles. We adopt SimCLR-style contrastive learning for WSI tiles.

However, there is an additional aspect unique to pathology that does not necessarily apply to natural images typically used in SimCLR applications. Histopathology necessitates local invariance—the diagnostic or prognostic information contained in a sampled tile should be similar to adjacent or overlapping tiles, and thus the learned representations should be similar. This is because microanatomy is localized—the neighbor of a random tile within a specific microanatomy (stroma, tumor, inflammation, etc.) is *likely* to be a part of the same anatomy and likely contains the same information. For example, if a pathologist makes an assessment such as counting positive cells in a high-power field, shifting that field over a few hundred microns should not make a significant difference in terms of diagnosis or prognosis, if any. We note that we are not the first to recognize this aspect of pathology with respect to self-supervision [27], although it was independently incepted. Thus, in addition to the transformation invariances implemented by SimCLR, we propose locality invariance. This yields a family of transformations—rotation, adjacency, and scale—which the proposed model learns to encode into an invariant representation. This will be the first way we will co-opt SimCLR in the proposed model—contrastive learning will be used to build representations of tiles that are invariant to transformed tiles. Figure 2 depicts this contrastive framework.

### 3.2. Multiple Instance Learning (MIL)

MIL [37,38] is a machine learning method where labels are assigned to collections (called bags) rather than individual examples (called instances), as in conventional machine learning. For histopathological analyses, bags consist of tessellated WSIs, in which each tile is a small unannotated image sampled from the WSI. It is popular for WSI analyses [4,17,19,20,86,87], as only slide-level labels are required for its training and implementation, thus negating the need for tissue-level annotations.

Arguably, the most successful adaptation of MIL within WSI analysis is attention-based (AB) MIL [17,18,19,20,21,86,87]. The attention-based pooling mechanism [88] automatically learns to dynamically weight embedded instances into a bag-level feature vector. A weight is automatically computed for each embedded instance, and then a weighted sum combines them into a single, bag-level instance corresponding to a slide-level embedding. Classification or regression is then performed on this bag-level embedding.
(1)$$a_{k}=\frac{exp\left\{w^{T}tanh\left(Vh_{k}^{T}\right)\cdot sigm\left(Uh_{k}^{T}\right)\right\}}{\sum_{j=1}^{K}exp\left\{w^{T}tanh\left(Vh_{j}^{T}\right)\cdot sigm\left(Uh_{k}^{T}\right)\right\}}$$
(2)$$z=\overset{K}{\underset{k=1}{\sum }}a_{k}h_{k}$$

Implementation of the attention mechanism consists of a simple two-layer fully connected network which passes each instance embedding (*h_k_*) through two parallel layers of the network (*V*, *U*). Then, it applies a tanh and sigmoid activation function to the respective results, dots the results, next passing the fused activation through another layer (*w^T^*), which maps the vector into a single value, its attention weight (*a_k_*). Equation (1) summarizes these interactions. The weighted sum of each embedded instance and its attention weight yields a bag-level instance (*z*), as in Equation (2). The parameters (*V*, *U*, *w*) for this two-layer neural network are automatically learned through model training.

### 3.3. Contrastive MIL

Contrastive learning is also applied on the MIL level. Commonly, MIL is utilized in a supervised manner to predict a label associated with the bag. For WSIs, this can be a diagnostic outcome. Inspired by SimCLR, we propose a novel method to learn effective representations of WSIs without supervision, using contrastive learning with MIL. With a transformation-invariant tile encoder trained as in Figure 2, pairs of bags are subsampled from the resulting tile encodings. These MIL bags serve as “transformations” of the WSI, similar to the color, scale, and rotation transformations of SimCLR. AB-MIL is used to aggregate MIL bags into bag-level encodings and similar to SimCLR, project via a projection head. Contrastive loss is then used to attract bag-level projections from the same slide and to contrast bag-level projections from different slides. With this framework, a label-free slide-level feature vector is created. These steps are summarized in Figure 3.

Our hypothesis with this latter framework is that the learned bag-level encodings of each slide contain features that are unique to each slide. Usually, in attention-based MIL, attention weights bias the bag-level encodings towards instances that correlate to the slide’s label. For example, when attention-based MIL is trained on a set of WSIs labeled for tumor presence or non-presence, tiles within the tumor receive higher attention [17,18]. This is what gives the model a degree of interpretability. However, for the proposed model, there are no labels initially. *Instead, the objective is to force differently sampled MIL bags of the same slide to map to the same bag-level encoding and make these bag-level encodings as different as possible from bag-level encodings of other slides.* To satisfy these constraints, we hypothesize that microanatomies that discriminate from one another are attended to. For example, assume we have two slides—both share two kinds of microanatomies, but one has an additional microanatomy. To maximize the contrast between these two slides, the bag-level encoding should be biased towards the unique microanatomy, which is not shared between the two slides. This bias (or weight) towards unique microanatomies (or tiles) is learned by SS-MIL. Ultimately, we hypothesize that the features learned by SS-MIL in bag-level encodings correspond to features of the slide with the maximum variance across the dataset.

### 3.4. Datasets

We utilize two publicly available datasets to test the proposed method. TCGA-NSCLC is a dataset of 1053 non-small-cell lung cancer H&E WSIs. The task is to subtype these biopsies into LUAD(*n* = 541) and LUSC (*n* = 512). This dataset serves as a “classification” prototype. The second dataset is TUPAC16 [89], which consists of 500 breast cancer H&E WSIs. The task is to predict a continuous proliferation score, reflecting the degree of mitotic activity within the tissue. This serves as a “regression” prototype. All slides are resampled to 20× magnification, regardless of the data source.

### 3.5. Experimental Design

A five-fold cross-validation was applied to each dataset. The proportion of training, validation, and testing slides was 60%, 20%, and 20%, respectively, in which each testing set is mutually exclusive from all other testing sets. These folds were kept consistent through each stage of model training and for comparison methods.

SimCLR pre-training was conducted using ResNet50 as a backbone with standard scaling, color jittering, and contrast jittering augmentations. Additionally, a separate set of experiments were carried out in which random neighboring tiles (i.e., those within one step of the anchor tile) were selected as augmentations of the anchor tile. Learning rate, optimizer, batch size, and projected feature dimensions were 0.001, Adam, 0.5, and 128, respectively. Typically, SimCLR is trained for hundreds of epochs without subsampling the data. However, recent studies have reported that increasing the training time beyond 20 epochs and exhaustively sampling all slides does not improve the resulting discriminative power of tile embeddings for histopathology images [23,60]. Therefore, our models were trained for 20 epochs and stopped if validation loss did not decrease for 5 epochs. 1000 tiles were randomly selected from each slide for pre-training. However, all tiles were passed through the resulting self-supervised encoder during inference.

For SS-MIL, models were trained using SimCLR embeddings with a learning rate of 0.0002, Adam optimizer with a weight decay of 10^−5^, and a batch size of 70. For positive pair generation, each bag was subsampled (with replacement) for one-quarter of the total instances in the bag. Contrastive loss during this step utilized a temperature parameter of 1.0. Models were trained for 1000 epochs, and the model with the lowest validation loss was saved. A separate set of SimCLR models were trained with neighboring tiles as additional augmentation. As in recent studies comparing instance-level self-supervised methods, we (1) fine-tune the resulting contrastive MIL model with a new fully-connected layer along with slide-level labels. Then, we (2) freeze the resulting contrastively MIL model and train a new fully connected layer along with slide-level labels, and (3) perform an ablation study in which only 25%, 50%, and 75% of the labels are utilized for both (1) and (2). Finally, we (4) compare with fully-supervised attention-based MIL, CLAM [17] and Attention2majority [20], using both SimCLR embeddings as well as generic ImageNet embeddings derived from a pre-trained ResNet50. For Attention2majority, we test different numbers of instances, as in the original study. For TCGA-NSCLC, we report the resulting accuracy for each class and AUC, and for TUPAC, we report the Pearson correlation. We refer to self-supervision via SimCLR as “SSL” and self-supervision via SimCLR with neighboring tiles as additional augmentation as “SSLn” in the Results.

## 4. Results

### 4.1. NSCLC Subtyping

Table 1 reports the results of the NSCLC subtyping task. Based solely on AUC, the highest performance was achieved using CLAM and ImageNet features, with an AUC of 0.9434 ± 0.0140. However, nearly the same performance was achieved when utilizing plain AB-MIL with ImageNet features, with an AUC of 0.9415 ± 0.0130. Furthermore, ImageNet features outperformed the same methods with SimCLR-derived features, with around a 0.02 decrease in AUC. However, these SimCLR-derived features did exhibit more balance between sensitivity and specificity comparatively. Furthermore, AB-MIL seems to perform better when utilizing SSL or SSLn features. As for Attention2majority, we again observed a slight decrease in overall performance, especially specificity. As expected, the more instances included in each bag, the higher the overall performance, regardless of feature set. This is consistent with previous findings [20]. Finally, SS-MIL achieved an AUC of 0.8641 ± 0.0115 and 0.8212 ± 0.0129 for respective SimCLR-derived and modified SimCLR-derived features.

### 4.2. TUPAC Proliferation Scoring

Table 1 also reports the results of the TUPAC proliferation scoring task. The highest R^2^ was 0.6790 ± 0.1108, achieved using AB-MIL with SimCLR-derived features and closely followed by the same method with ImageNet features with R^2^ of 0.6738 ± 0.0432. However, we noted that though the average R^2^ was slightly higher for SimCLR-derived features, a much larger standard deviation was observed, perhaps suggesting that ImageNet features are superior. Moving on, a stark drop-off in performance was observed when utilizing modified SimCLR-derived features, with an R^2^ of 0.5776 ± 0.1075. Like the NSCLC subtyping task, Attention2majority performance increased when the number of instances increased. Likewise, SS-MIL did not achieve the same level of performance as the supervised counterparts, achieving an R^2^ of 0.5764 ± 0.0996 and 0.4018 ± 0.0828 for respective SimCLR-derived and modified SimCLR-derived features.

### 4.3. Ablation Studies

Table 2 reports the result of the NSCLC and TUPAC ablation studies. Overall, we observed an increase in performance with more slides in the training set, as previously reported in other studies [17,20,23]. As in the results reported in Table 1, SimCLR-derived features outperformed respective SimCLR-derived features with neighboring tiles utilized as an additional augmentation. For the NSCLC subtyping task, supervision with AB-MIL clearly exceeded the performance of fine-tuning SS-MIL, which in turn exceeds the performance of the proposed model. In each case, we observed a decrease in AUC of 0.02 and 0.04 relative to 75% when 50% and 25% of the slides were used, respectively. For the TUPAC proliferation scoring task, fine-tuning SS-MIL outperformed supervised attention-based MIL, which outperformed SS-MIL, unlike the NSCLC subtyping task.

## 5. Discussion

### 5.1. SS-MIL Can Leverage Completely Unannotated Datasets

The lack of need for labels, tissue-level or slide-level, opens up several opportunities for research that are simply not possible with existing methods. SS-MIL allows for datasets to be combined even though they may have different kinds of slide-level labels *or* if no slide-level labels exist for a particular data source (i.e., missing data). For example, the TUPAC16 dataset could be combined with the TCGA-BRCA (breast invasive carcinoma). Our proposed model could be trained on TCGA-BRCA and then either frozen or fine-tuned on TUPAC16, or vice-versa. Furthermore, SS-MIL could be trained on both TCGA-BRCA and TUPAC16 and then frozen or fine-tuned on a smaller, external dataset. The benefit SS-MIL derives from combining different datasets to learn slide-level representation is impossible with existing methods. It is not just publicly available datasets that could theoretically be combined, and benefit derived from. Hospitals and clinical trials have large swaths of WSIs which are easily accessible, but their slide-level labels are either prohibitively difficult to obtain, irrelevant to the problem of interest, or non-existent [90,91].

### 5.2. SS-MIL Still Underperforms Compared to Supervised Methods

Overall, SS-MIL’s performance is not on par with supervised methods’ performance in both tasks. However, this is to be expected. All supervised comparison—attention-based MIL, CLAM, and Attention2majority—can learn slide-level feature spaces that can more easily discriminate NSCLC subtypes or regress proliferation scores. Furthermore, each of these methods, whether utilizing SimCLR-derived features or ImageNet features, partly consist of a shallow feature extractor, allowing the network to learn a tile-level feature space which ultimately contributes to a more discriminable slide-level feature space. By comparison, SS-MIL does not benefit from these advantages. Instead, it entirely relies on the target of transformation invariance to learn meaningful tile-level features. Likewise, SS-MIL relies on the power of contrastive learning and attention to learn a sampling-invariant representation of each slide rather than a label-dependent feature space.

This is evidenced by three observations. Firstly, fine-tuning SS-MIL clearly outperforms itself when training a new linear layer on the unsupervised slide-level representations. With fine-tuning, only the starting point of the network is different, and slide-level feature space dependent on the slide-level labels is more easily achieved. Secondly, training any of the comparison methods using SimCLR-derived features and slide-level labels outperforms fine-tuning. In other words, each supervised method can learn a favorable tile-level feature space in which slide-level representations are more easily distinguished. Thirdly, during model training, we observed that all supervised methods always yielded training set losses approaching zero. Of course, this did not result in model overfitting, as early stopping criteria was based on validation loss. However, this behavior was in strong contrast to SS-MIL. No matter how long SS-MIL was trained, training loss always converged to a level well-above zero and nearly matched validation and testing losses. This observation supports the notion that supervised comparisons learn a slide-level feature space that benefits greatly from labels. We also believe this observation indicates that SS-MIL is self-regularizing to a degree. In summary, there are several benefits to supervised methods that SS-MIL does not benefit from.

Despite the advantages of supervision, the results of the ablation study offer some respite. The results suggest that SS-MIL still learns a robust feature space. This is evidenced by the observation that the decrease in performance when utilizing fewer slides for SS-MIL is about the same as comparable methods in the ablation study (i.e., fine-tuning and supervision). In addition, evidence is derived from the apparent maintenance of performance despite the decrease in the number of training slides.

### 5.3. Generic Features Outperform Histopathology Features

The results also suggest that ImageNet features may be advantageous over histopathology-specific features. Several studies have sought to utilize histopathology-specific tile features for WSI analysis either using self-supervision [21,22,24,25,27], weak supervision using slide-level labels [20,65], supervision on unrelated histopathology tasks [29,92], in an end-to-end manner [86,87], or graph-based methods [93]. The argument made by such studies is that histopathology WSI analysis should utilize histopathology-specific tile features. Without context, this makes sense. However, this point-of-view is entirely upended by the vast body of work across medical image analysis which utilizes transfer learning with small image datasets [94]. Furthermore, and by contrast, several studies use generic ImageNet features [17,18,19] for WSI analysis with great success. These studies and the results presented here call into question the necessity of histopathology-specific features for WSI analysis. Be that as it may, we do not suggest rejecting the prospect of histopathology-specific features altogether, as studies have shown that self-supervision as a means of reducing label load is far superior to transfer learning from ImageNet features [60,61].

### 5.4. Neighbors as Augmentations Does Not Benefit Downstream MIL

Contrary to supporting previous studies’ utilization of ImageNet features, our results do not support previous findings that consider neighboring tiles as augmentations for self-supervision to result in better representations than standard augmentations. In SimTriplet [27], the authors extended SimSiam [17] to include an additional stop-gradient arm that operated solely on encoded neighboring patches. The loss was then computed between the original SimSiam augmentation and the anchoring tile as well as between the neighbor tile and anchor tile. The idea here is that not only should tiles be invariant to their transformation but that they are nearly identical to their neighbors. Their method outperformed SimSiam on a melanoma histopathology tile classification task. We also independently recognized this aspect of histopathology that may be exploited by self-supervision. However, in all our experiments, SLLn features (with neighbors considered as augmentations) always underperformed compared to SSL features (not considering neighbors). This contradicts the results presented in SimTriplet [27]. We believe that the gap between these two approaches could be narrowed and flipped if our SimCLR encoder had ample time to learn SLLn representations. This is supported by our observation that the contrastive loss for the SLLn SimCLR encoder was higher at stopping time compared to the SLL SimCLR encoder (average 3% higher). We did not let either encoder train longer because previous studies showed that this does not affect downstream patch-wise classification accuracy [23,60]. However, these studies did not consider neighbors as augmentations nor were they performing slide-level classification or regression. Perhaps under these conditions, training the encoder longer may improve slide-level performance.

### 5.5. Why CLAM Outperforms Attention2majority

Contrary to expectations, CLAM outperformed Attention2majority in the NSCLC subtyping task. In a previous study [20], Attention2majority outperformed CLAM in a multi-class kidney grading task and Camelyon16. We believe this is due to the difference between the tasks. In NSCLC subtyping and TUPAC proliferation scoring, the signal AB-MIL attends to is well-dispersed throughout the tissue. In contrast, in Camelyon16, the signal may be a very small lesion, even smaller than the size of a tile, and in the kidney grading task, the signal is mixed. Attention2majority outperforms CLAM on these latter datasets because of its intelligent sampling scheme, which verifiably biases bags to include tiles that correspond to the overall slide-level label. CLAM utilizes all tiles, so the signal becomes harder to detect. Given that the signal is more dispersed in the NSCLC and TUPAC datasets, this effect is less pronounced, so the advantages afforded by Attention2majority no longer apply.

### 5.6. Issues with Reproducibility

Contrary to our expectations, we could not reproduce the results reported by CLAM on the NSCLC subtyping task. In the original paper, CLAM achieved an AUC of 0.9561 ± 0.0179. Granted, we did not have the same experimental setup regarding slide divisions among folds. However, other studies have been unable to reproduce the results reported by CLAM for NSCLC subtyping. TransMIL reported 0.9377 [25], neural image compression reported 0.9477 [95], and node-aligned graphs reported 0.9320 [96]. Similarly, contrary to our expectations, AB-MIL performed as well as CLAM with ImageNet features and performed better when utilizing SSL or SSLn features. This is corroborated by one other study [96]. These observations highlight the importance of reproducibility in deep learning methodologies.

### 5.7. Areas for Improvement

SS-MIL has many areas for improvement. First and foremost, given the results presented in the ablation study, a larger and label-free WSI dataset would likely be beneficial. Clearly, increasing the number of slides at the level of constative MIL improves downstream performance (Table 2). Similar benefits from more slides may also be derived for the initial tile-level SimCLR pretraining step, as is supported by one study [60]. However, SS-MIL would not require slide-level labels from a larger dataset, unlike previous methods.

Notwithstanding, such datasets would not necessarily be beneficial if it were not the same tissue type or disease category [23]. The proposed method may benefit from longer training times for the SimCLR encoder (specifically when neighbors are used as augmentations). We may also consider a more apt augmentation space. The original SimCLR augmentation space allows for two random crops from the same image to serve as a positive pair. We can modify this random cropping augmentation in two ways. First, our experiments are performed at 20×. When randomly cropping a 20× tile, the resulting crop is unsampled. However, since we have access to the WSI, we may instead grab the crop at 40× and then down sample it. In this manner, the tile crop augmentations contain more detail.

Secondly, we may consider a random center crop as an augmentation rather than a random crop. In other words, a tile is randomly cropped by keeping the center position the same and then resampled directly from the slide from 40×. This is motivated by the observation that two random crops from the same tile may not be adjacent. However, by center cropping, we can be sure that they overlap. Ultimately, augmentation space greatly affects downstream performance and may even be dataset-dependent [23,27]. The modifications that could be made with respect to augmentation space are many [97]. We could also modify our positive/negative pair generation during contrastive MIL model training. Currently, different bag views are generated by randomly subsampling 25% of the WSI tile embeddings.

However, for tasks in which the region of interest is very small, such as metastasis detection, this sampling method will miss regions of interest and thus generate positive pairs in which one bag contains the region of interest and the other does not. Alternatively, we could generate multiple bags for the same slide by shifting the starting position of the tile grid. This way, each bag contains all tiles from the WSI but is also distinct from other bags from the same WSI. Additional augmentation policies could be applied to WSI bags, including random perturbation and random zeroing [98].

### 5.8. Implications of SS-MIL

SS-MIL enables researchers to benefit from a dataset where no label information is available. Its innovation lies 1) in the subtly of histopathology that neighboring tissue structures are likely to represent relatively the same (or very similar) clinical information and thus should be represented similarly as embeddings and 2) in the unsupervised nature of the proposed model. As the learned WSI embeddings have general applicability to many machine learning tasks (classification and regression), many other applications can benefit from these results.

The lack of the need for labels opens several opportunities for research that are simply not possible with existing methods. SS-MIL allows for datasets to be combined even though they may have different kinds of slide-level labels or if no slide-level labels exist for a particular data source (i.e., missing data). For example, the TUPAC16 dataset could be combined with the TCGA-BRCA (breast invasive carcinoma). Our proposed model could be trained on TCGA-BRCA and then either frozen or fine-tuned on TUPAC16, or visa-versa. Furthermore, SS-MIL could be trained on both TCGA-BRCA and TUPAC16 and then frozen or fine-tuned on a smaller, external dataset. The benefit SS-MIL derives from combining different datasets to learn slide-level representation is impossible with existing methods. It is not just publicly available datasets that could theoretically be combined, and benefit derived from. Hospitals and clinical trials have large swaths of WSIs which are easily accessible, but their slide-level labels are either prohibitively difficult to obtain, irrelevant to the problem of interest, or non-existent.

Furthermore, we have yet to examine the attention weights from the SS-MIL. We hypothesized that the features learned in bag-level encodings correspond to features of the slide with the maximum variance across the dataset and that instances with the highest attention would correspond to these histopathological features. It may prove difficult to support such a hypothesis with this dataset. However, perhaps with a hand-crafted dataset with a diversity of tissue structures (such as one WSI per organ), we may indeed demonstrate that the SS-MIL method possesses learning imaging features with the highest variance across a given dataset.

## 6. Conclusions

In conclusion, we have presented a method to learn compact representations of WSIs without supervision. Our method trains a patch-wise encoder using SimCLR. Each patch is embedded to yield a collection of instances for each WSI. Each collection is divided into several subsets, each representing the WSI, which MIL then fuses to yield multiple slide-level representations. Through this operation on the same slide, a positive pair for contrastive learning is created. Similarly, a negative pair is created for a different slide. This MIL model is then trained using contrastive loss. These unsupervised representations can be utilized to classify and regress WSIs to weakly labeled outcomes. We applied our method to both NSCLC subtyping and TUPAC proliferation scoring, achieving an AUC of 0.8641 ± 0.0115 and R^2^ of 0.5740 ± 0.0970. Though our method does not achieve the same level of performance as supervised attention-based MIL, CLAM, or Attention2majority, we have shown through ablation experiments that the slide-level feature space that its learning is robust and its performance is likely limited solely by the number of available slides. In future experiments, we plan to apply our method to larger datasets in order to observe whether the apparent benefits from increasing the number of slide (as evidence in ablation studies) continues to return benefits in performance. We expect that the performance of our method may indeed exceed that of supervised methods when limited labels are available. Secondly, we plan to modify our tile-level augmentation space to more accurately reflect the histopathology-specific transformation invariance (i.e., center crops). We will also perform separate experiments to find an optimal transformation policy. Similarly, we plan to modify our slide-level augmentation space (via shifting the overlaid grid or random zeroing) to represent each slide fully rather than randomly subsampling as in the current study. Thirdly, we will apply our resulting method to Camelyon16, a breast cancer seminal lymph node metastasis dataset, in conjunction with the novel, in-house MIL models. From a technical standpoint, our proposed method is a novel approach to computational pathology, where meaningful features can be learned from WSIs without needing any annotations. The proposed method can benefit computational pathology from a practical standpoint, as it would theoretically enable researchers to benefit from vast amounts of unlabeled or irrelevantly labeled.

## Figures and Tables

**Figure 1 cancers-14-05778-f001:**
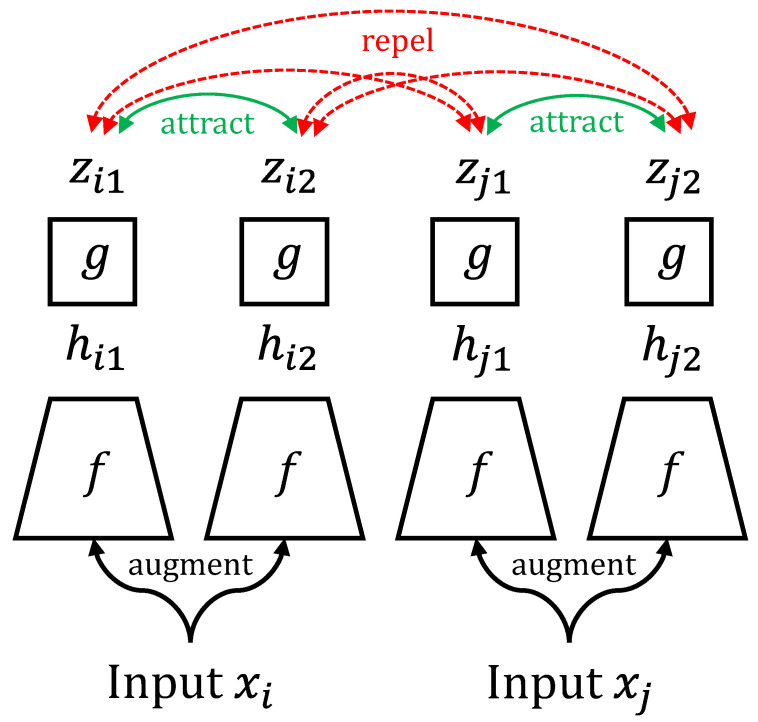
Contrastive learning as presented in SimCLR [39]. The input images *x_i_* and *x_j_* are augmented via random transformations, embedded via encoder *f* to yield embedding *h*, and then projected via *g* to yield projection *z*. The contrastive objective maximizes the similarity between positive pair *z_i_*_1_ and *z_i_*_2_ as well as positive pair *z_j_*_1_ and *z_j_*_2_ (i.e., “attract”) while minimizing the similarity between all other (negative) pairs (i.e., repel).

**Figure 2 cancers-14-05778-f002:**
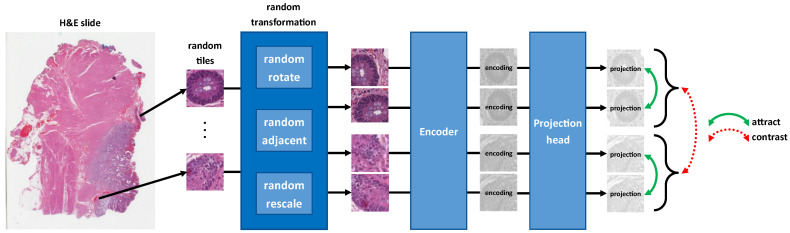
SimCLR learns effective representations of tiles from WSIs, which are invariant to rotation, scale, and adjacency. Each input tile is subject to two random transformations, both composed of a random rotation, selection of an adjacent tile, and random rescaling. This yields two transformed tiles for each input tile. All transformed tiles are encoded via a feature extractor, projected via a fully connected layer, and finally contrasted with transformations of all other input tiles and attracted to transformations of itself.

**Figure 3 cancers-14-05778-f003:**
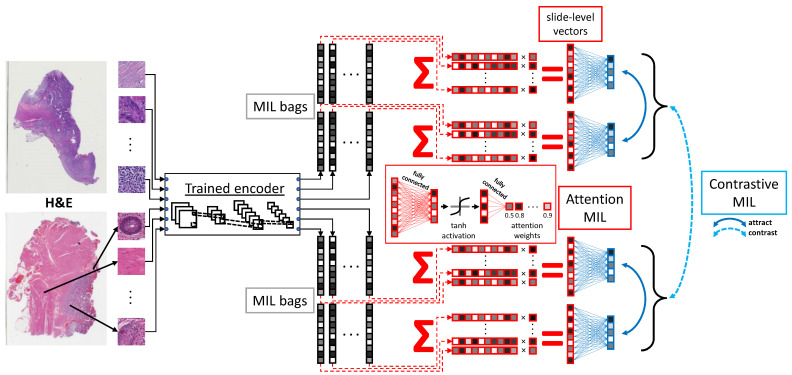
Encodings learned as in Figure 2 are utilized to construct multiple MIL bags per WSI. Each bag is aggregated into a bag-level encoding via attention-based MIL, trained via contrastive learning. Here, each bag-level encoding from a given slide acts as a “transformation” and attracts each other while contrasting with bag-level encoding from other slides. Note that the example here is only for one training iteration. Each iteration yields different pairs of MIL bags for each slide.

**Table 1 cancers-14-05778-t001:** NSCLC subtyping sensitivity, specificity, and the AUC and TUPAC proliferation scoring correlation (R^2^) for various methods: AB-MIL [88], CLAM [17], Attention2majority [20] (with increasing number of instances), and the proposed method. SSL refers to self-supervision via SimCLR, while SSLn refers to self-supervision via SimCLR with neighboring tiles as additional augmentation.

	NSCLC			TUPAC
Method	Sensitivity	Specificity	AUC	R^2^
AB-MIL				
ImageNet features	0.8528 ± 0.0323	**0.8779 ± 0.0330**	0.9415 ± 0.0130	0.6738 ± 0.0432
SSL features	0.8473 ± 0.0279	0.8461 ± 0.0352	0.9259 ± 0.0188	**0.6790 ± 0.1108**
SSLn features	0.7465 ± 0.0711	0.8157 ± 0.0453	0.8674 ± 0.0246	0.5776 ± 0.1075
CLAM				
ImageNet features	**0.8800 ± 0.0376**	0.8622 ± 0.0419	**0.9434 ± 0.0140**	-
SSL features	0.8427 ± 0.0430	0.8407 ± 0.0171	0.9240 ± 0.0147	-
SSLn features	0.7373 ± 0.0503	0.7879 ± 0.0779	0.8479 ± 0.0553	-
Attention2majority				
SSL features				
100 (instances)	0.8047 ± 0.0707	0.7406 ± 0.0843	0.8504 ± 0.0177	0.5518 ± 0.0567
200	0.8002 ± 0.0463	0.7525 ± 0.0731	0.8651 ± 0.0284	0.5943 ± 0.0471
500	0.8214 ± 0.0374	0.7663 ± 0.0755	0.8719 ± 0.0244	0.6218 ± 0.0892
1000	0.8480 ± 0.0180	0.7565 ± 0.0652	0.8770 ± 0.0119	0.6361 ± 0.0855
2000	0.8486 ± 0.0284	0.7649 ± 0.0599	0.8848 ± 0.0151	0.6431 ± 0.0783
5000	0.8609 ± 0.0296	0.7455 ± 0.0230	0.8916 ± 0.0192	0.6492 ± 0.0877
10,000	0.8708 ± 0.0103	0.7237 ± 0.0349	0.8931 ± 0.0152	0.6505 ± 0.0879
SSLn features				
100 (instances)	0.7708 ± 0.0822	0.7394 ± 0.0759	0.8246 ± 0.0453	0.4161 ± 0.1163
200	0.7730 ± 0.0920	0.7573 ± 0.0842	0.8347 ± 0.0320	0.4422 ± 0.0872
500	0.7672 ± 0.1049	0.7465 ± 0.0959	0.8408 ± 0.0289	0.5308 ± 0.0586
1000	0.7687 ± 0.0979	0.7628 ± 0.0648	0.8425 ± 0.0403	0.4888 ± 0.0655
2000	0.7811 ± 0.0907	0.7576 ± 0.0401	0.8452 ± 0.0352	0.5192 ± 0.0551
5000	0.8092 ± 0.0983	0.7280 ± 0.0983	0.8478 ± 0.0329	0.5066 ± 0.0889
10,000	0.8208 ± 0.0857	0.7334 ± 0.0620	0.8490 ± 0.0237	0.4718 ± 0.1066
SS-MIL (proposed)				
ImageNet	0.7363 ± 0.0315	0.7533 ± 0.0521	0.8245 ± 0.0392	0.5418 ± 0.0330
SSL features	0.8720 ± 0.0361	0.7888 ± 0.0336	0.8641 ± 0.0115	0.5740 ± 0.0970
SSLn features	0.8251 ± 0.0509	0.7598 ± 0.0328	0.8212 ± 0.0129	0.4611 ± 0.2058

**Table 2 cancers-14-05778-t002:** NSCLC and TUPAC ablation study. Supervised refers to supervised attention-based MIL from scratch. Fine-tune refers to fine-tuning SS-MIL model. Frozen refers to training only a new linear layer after training the SS-MIL. SSL refers to self-supervision via SimCLR, while SSLn refers to self-supervision via SimCLR with neighboring tiles as additional augmentation.

	NSCLC			TUPAC
Method	Sensitivity	Specificity	AUC	R^2^
SSL features				
Supervised				
25%	0.7898 ± 0.0967	0.7001 ± 0.1220	0.8204 ± 0.0521	0.4214 ± 0.1853
50%	0.8059 ± 0.0769	0.7672 ± 0.0747	0.8633 ± 0.0302	0.4680 ± 0.1848
75%	0.8268 ± 0.0537	0.7795 ± 0.0705	0.8858 ± 0.0212	0.5453 ± 0.0711
Fine-tune				
25%	0.7951 ± 0.0796	0.6691 ± 0.1165	0.8081 ± 0.0602	0.4898 ± 0.0841
50%	0.8153 ± 0.0655	0.7336 ± 0.0661	0.8545 ± 0.0277	0.5320 ± 0.0936
75%	0.8537 ± 0.0670	0.7260 ± 0.0837	0.8744 ± 0.0175	0.5692 ± 0.0831
Frozen				
25%	0.7176 ± 0.0932	0.6784 ± 0.0937	0.7787 ± 0.0375	0.3912 ± 0.0940
50%	0.7651 ± 0.0355	0.7597 ± 0.0477	0.8358 ± 0.0249	0.4289 ± 0.1012
75%	0.7721 ± 0.0405	0.7696 ± 0.0452	0.8532 ± 0.0138	0.4283 ± 0.0966
SSLn features				
Supervised				
25%	0.7453 ± 0.1107	0.4948 ± 0.1320	0.6749 ± 0.0534	0.3311 ± 0.0976
50%	0.7236 ± 0.0960	0.6139 ± 0.1707	0.7465 ± 0.0585	0.4059 ± 0.0914
75%	0.7881 ± 0.0748	0.6331 ± 0.1448	0.7871 ± 0.0581	0.4603 ± 0.0925
Fine-tune				
25%	0.6627 ± 0.3339	0.5047 ± 0.3551	0.7079 ± 0.0574	0.3950 ± 0.1203
50%	0.8239 ± 0.1157	0.5140 ± 0.2054	0.7690 ± 0.0543	0.4588 ± 0.0957
75%	0.8385 ± 0.0779	0.5306 ± 0.2163	0.7916 ± 0.0572	0.5078 ± 0.1040
Frozen				
25%	0.6981 ± 0.0765	0.6740 ± 0.0658	0.7461 ± 0.0293	0.3731 ± 0.1376
50%	0.7106 ± 0.0503	0.7153 ± 0.0686	0.7846 ± 0.0264	0.4160 ± 0.1537
75%	0.7418 ± 0.0447	0.7373 ± 0.0460	0.8097 ± 0.0235	0.4547 ± 0.1608
ImageNet				
Supervised				
25%	0.8112 ± 0.0725	0.8123 ± 0.0696	0.8958 ± 0.0150	0.5670 ± 0.0672
50%	0.8261 ± 0.0679	0.8569 ± 0.0501	0.9197 ± 0.0153	0.6291 ± 0.0627
75%	0.8602 ± 0.0664	0.8537 ± 0.0408	0.9288 ± 0.0117	0.6446 ± 0.0385
Fine-tune				
25%	0.7776 ± 0.0944	0.8188 ± 0.0788	0.8863 ± 0.0227	0.5511 ± 0.0699
50%	0.8011 ± 0.0655	0.8743 ± 0.0512	0.9196 ± 0.0173	0.5986 ± 0.0523
75%	0.8245 ± 0.0684	0.8649 ± 0.0510	0.9257 ± 0.0118	0.6297 ± 0.0455
Frozen				
25%	0.7065 ± 0.0889	0.6512 ± 0.0975	0.7324 ± 0.0668	0.4827 ± 0.0552
50%	0.7497 ± 0.0471	0.7383 ± 0.0660	0.8205 ± 0.0329	0.5072 ± 0.0696
75%	0.7663 ± 0.0365	0.7474 ± 0.0470	0.8386 ± 0.0256	0.5222 ± 0.0503

## Data Availability

TCGA-NSCLC slides are available from the GDC data portal (https://portal.gdc.cancer.gov/). TUPAC16 slides are available from the grand challenge website (https://tupac.grand-challenge.org/). Code will be available at https://github.com/cialab/SSMIL.

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
