# Peer review of "Contrastive Multiple Instance Learning: An Unsupervised Framework for Learning Slide-Level Representations of Whole Slide Histopathology Images without Labels"

_cancers, 2022, doi:10.3390/cancers14235778_

Round 1
Reviewer 1 Report
This paper presents a technique to learn meaningful features from WSIa without the need for slide-level annotations based on contrastive learning. The presentation of motivations and methods is clear and the results are well discussed, providing very interesting insights.
While I believe that the paper is already well written and well structured, the clarity and self-containment of the manuscript would majorly benefit from a more in-depth discussion of the background and related works, than the one that is now briefly reported in the Introduction. I suggest the authors to provide a specific section for that, before methods are described.
Reviewer 2 Report
The paper is generally well-written, and the idea is effectively conveyed. This paper needs careful revision. The authors should revise the work by considering the reviewer's recommendations.
In the introduction, what key theoretical perspectives and empirical findings in the main literature have already informed the problem formulation? What major, unaddressed puzzle, controversy, or paradox does this research address?
Authors should further clarify and elaborate novelty in their contribution.
A few abbreviations are not elucidated on the first appearance. Moreover, the initial letters for each acronym must be consistent (All capital is preferable, in my opinion).
Authors focus on article references, though they are many good recently published books and chapters that can be cited.
Below papers have some interesting implications that you could discuss in your introduction and how it relates to your work.
Vulli, A.; et al.. Fine-Tuned DenseNet-169 for Breast Cancer Metastasis Prediction Using FastAI and 1-Cycle Policy. Sensors 2022, 22, 2988.
El-Sappagh, Shaker, et al. "Automatic detection of Alzheimer’s disease progression: An efficient information fusion approach with heterogeneous ensemble classifiers." Neurocomputing (2022).
The conclusion section is concise. There is a need to work on it further.
Future work is better added to the conclusion section.
Improve figures quality by using vector format figures
What are the limitations of the present work?
What are the practical implications of this research?
Round 2
Reviewer 2 Report
All comments are addressed.